# Put Your Money Where Your Mouth Is: Evaluating Strategic Planning and Execution of LLM Agents in an Auction Arena

## Abstract

*Can Large Language Models (LLMs) simulate human behavior in complex environments?* LLMs have recently been shown to exhibit advanced reasoning skills but much of NLP evaluation still relies on static benchmarks. Answering this requires evaluation environments that probe strategic reasoning in competitive, dynamic scenarios that involve long-term planning. We introduce AucArena, a novel simulation environment for evaluating LLMs within auctions, a setting chosen for being highly unpredictable and involving many skills related to resource and risk management, while being easy to evaluate. We conduct several controlled simulations using state-of-the-art LLMs as bidding agents. We find that through simple prompting, LLMs do indeed demonstrate many of the skills needed for effectively engaging in auctions (e.g., managing budget, adhering to long-term goals and priorities), skills that we find can be sharpened by explicitly encouraging models to be adaptive and observe strategies in past auctions. These results are significant as they show the potential of using LLM agents to model intricate social dynamics, especially in competitive settings. However, we also observe considerable variability in the capabilities of individual LLMs. Notably, even our most advanced models (GPT-4) are occasionally surpassed by heuristic baselines and human agents, highlighting the potential for further improvements in the design of LLM agents and the important role that our simulation environment can play in further testing and refining agent architectures. Our resources will be released.

## 1 Introduction

A long-term goal of the AI community has been the development of autonomous agents that can independently make decisions and freely interact in the environment to carry out different tasks (Steels, 1995; Franklin & Graesser, 1996). Being autonomous requires an agent to have a certain set of skills, such as the ability to do complex reasoning, and manage risk and resources, among many others. Large Language Models (LLMs) have proven to be able to solve a wide range of different reasoning problems, with the boundaries of what's possible being pushed every day (Wei et al., 2022a; Bubeck et al., 2023). Despite the increasing view of these models as autonomous agents (Wang et al., 2023a; Sumers et al., 2023; Xi et al., 2023), a crucial question remains: *Can these agents effectively do sequential decision-making in dynamic environments for achieving their strategic objectives?*

While the potential is evident (Nakajima, 2023; Significant-Gravitas, 2023), these capabilities have yet to be rigorously evaluated. Traditional reasoning and planning benchmarks in NLP (Geva et al., 2021; Sakaguchi et al., 2021; Yuan et al., 2023) mostly assess agents in static contexts. Yet, real-world scenarios demand that autonomous agents not merely respond to input but also have the ability to create long-term goals and plans, and continuously revise their decisions. To bridge this gap, one recent line of research focuses on immersing agents in simulation environments that mimic real-world scenarios (Wang et al., 2022; Park et al., 2023; Liu et al., 2023), ones that often focus on a targeted set of skills. However, designing such simulations can require significant engineering effort, and doing fine-grained evaluation in these environments can often be a challenge.

In this work, we focus on building simulated dynamic environments that require natural and interesting types of strategic reasoning, yet that are easy to build and evaluate. We emphasize developing

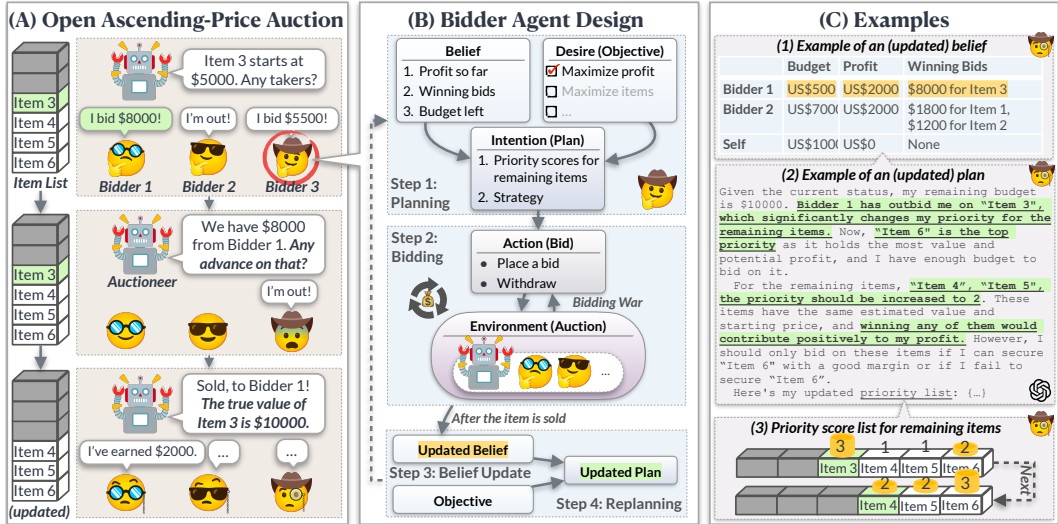

Figure 1: An illustration of AUCARENA. (A) demonstrates a multi-round, open ascending-price auction, with an auctioneer collecting bids from everyone and announcing the current highest bids and bidders making decisions publicly after some private reasoning. (B) depicts the bidder agent architecture based on the Belief-Desire-Intention Model, including four actions: planning, bidding, belief update, and replanning. Beliefs and plans are updated for every item based on the understanding of the auction status and driving desires. (C) provides examples of updated beliefs and plans, showing a bidder assigning priority scores to remaining items after intermediate reasoning.

environments characterized by the following properties: *1)* They are dynamic and inherently unpredictable, requiring agents to be adaptive; *2)* The environments involve limited resources, making the assets for competition scarce and the rewards highly contested; *3)* The actions that agents carry out are quantifiable, facilitating easy evaluation. Motivated by the dynamics of *auctions*, which are widely studied in multi-agent systems and game theory (Laffont, 1997; Tuyls & Parsons, 2007), we introduce AUCARENA. Auctions offer a fertile ground for assessing strategic planning, resource allocation, risk management, and competitive behaviors. In AUCARENA, agents act as bidders in simulated auctions, grounding their strategic capabilities into execution. This environment enables the quantification of an agent's performance using numerical metrics, such as total profit.

In the simulated environment (Figure 1), we set up an open auction where all bidders have equal information. Bidder agents engage in long-term sequential auctions with a fixed budget, allowing for strategic considerations. Each agent monitors the environment and forms a plan to achieve its objective. To facilitate interaction with the auction environment, we incorporate four essential functions: planning, bidding, belief update, and replanning, each implemented with LLM prompting (Ouyang et al., 2022; Wei et al., 2022b). These functions manifest the agents' strategic planning, adaptation to new information, and real-time decision-making capabilities. We systematically explore the individual and collective behaviors of various LLM agents, examine their plan execution and adaptability, and delve into the dynamics of multi-agent competition and interactions between humans and AI. Our contributions are: *1)* We present AUCARENA, a simulation environment crafted to assess the strategic planning and execution abilities of autonomous LLM agents in dynamic, competitive scenarios with limited assets; *2)* We develop a bidder agent architecture that enables LLMs to skillfully strategize execute, and modify their plans in response to the environment; *3)* Utilizing AUCARENA, we gain insights into the varied capabilities of LLMs, emphasizing adaptive planning's importance in competition. We uncover new dynamics of niche separation in diverse multi-agent scenarios, showing that even superior models like GPT-4 don't always prevail in long-term strategic alignment.

## 2 RELATED WORK

**Autonomous Agents** Research in AI has tried to clarify what it means to be autonomous, which loosely includes having environmental perception, goal-oriented actions, and the ability to self-

improve (McCarthy et al., 2006; Goodwin, 1995). AI agents have evolved from being symbolic in nature, utilizing production rules and semantic networks (Newell & Simon, 2007), to interactive, applying reinforcement learning and neural networks for enhanced responsiveness in dynamic environments (Mnih et al., 2013; Silver et al., 2016; 2017). The development of *generative LLM agents* has further broadened AI capabilities in natural language understanding and generation, unlocking new possibilities (Wang et al., 2022; Significant-Gravitas, 2023). In this work, we extend the concept of *generative agents* and use pre-trained LLMs as the foundational element.

**Generative LLM Agents**  Generative LLM agents (*i.e.*, generative LLMs as controller) rely on LLMs for adaptive autonomy, proficiency in task decomposition, and generalization in decision-making, execution, and learning (Wei et al., 2021; 2022b; Wang et al., 2023b). These agents have diverse applications in robotics, law, and research (Ahn et al., 2022; Cui et al., 2023; Bran et al., 2023). Multi-agent interactions yield insights into social simulation and cooperation in software development (Park et al., 2023; Qian et al., 2023). Unlike cooperative multi-agent scenarios, we situate our agents in a competitive auction game, examining their strategic planning and adaptability in dynamic, resource-constrained settings.

**Simulation-based Agent Evaluation of LLMs**  LLMs' remarkable versatility exposes limitations of traditional benchmark evaluations for specific NLP tasks (Wang et al., 2018; 2019; Geva et al., 2021; Sakaguchi et al., 2021), giving rise to the evaluation of LLM agents (Xi et al., 2023). SmallVille (Park et al., 2023) provides insights into how LLMs behave as agents in social simulation, but agent interactions lack clear objectives, making quantitative analysis challenging. Fu et al. (2023) evaluates LLMs' negotiation skills in bargaining using the deal price as the metric, but bargaining does not assess LLMs' long-term strategic planning abilities. AgentBench (Liu et al., 2023) is a comprehensive benchmark for evaluating LLM agents across 8 scenarios, but these tasks focus on single agents and are limited in terms of evaluating LLMs in dynamic environments. In our work, we leverage multi-agent auctions as a test-bed for assessing LLM agents' strategic planning, execution and adaptability in objective-driven environments with precise, measurable metrics.

**Agent-based Social Simulation**  In social sciences, there is a large literature on *agent-based social simulation* (Gilbert & Troitzsch, 2005), which employs a variety of formal techniques to model scenarios such as social dilemmas (Axelrod, 1997; Gotts et al., 2003) argumentation and opinion dynamics (Betz, 2012; Mäs & Flache, 2013) and economics (Hamill & Gilbert, 2015). Our work takes inspiration from this line of work, and the broader goal of using simulations to better understand social dynamics by studying artificially constructed scenarios that might otherwise be difficult or impossible to study in the real world. Our work, however, focuses on simulation using LLM-based agents, which have the promise of being a more natural fit for many agent simulation problems over traditional formal models due to their vast knowledge and their ability to easily converse in ordinary language (see discussion in Betz (2021)).

## 3  THE AUCTION ARENA

We introduce the implementation of our AUCARENA (§ 3.1) and agent architecture (§ 3.2) we use for our simulation. Real-life auctions can be complicated, but the fundamental principle of AUCARENA is to keep the environment simple and controllable, while being realistic as much as possible. Therefore, we will also introduce some details about our simulation design and implementation in § 3.3.

### 3.1  OPEN ASCENDING-PRICE AUCTION

In this work, we adopt the English Auction setting, *i.e.*, an auction where items are introduced in rounds by an auctioneer who, starting from an initial value, accepts increasingly higher bids (ascending-price) from participants until the highest bid is obtained, where all bidders' actions are transparent and observable for fairness. The primary rule is that the highest bid wins the item. Such a process is illustrated in Figure 1(A) and includes the following components: a list of items, an auctioneer, and bidders. An example auction log is provided in Appendix A.1.

**Items**  Items are central to the auction. We create a transparent environment that allows bidders to make informed decisions. Each item has a *starting price* (*e.g.*, $1,000) for a bidding war, and a *true value* ($2,000), which is common for everyone and practically the value for resale. For simplicity, we do not introduce additional values such as various personal preferences and emotional value of items.

**Auctioneer** To keep the auction simple and stable, the auctioneer is operationalized as a static symbolic agent. The auctioneer can be viewed as dictating the rules in the AUCARENA environment: as a neutral arbiter, the auctioneer manages the flow of the bidding process, declares the winning bid, and enforces auction rules, *e.g.*, bidders can not exceed their budgets, the next bid must surpass the prior highest bid by a minimum increase, etc.

**Bidders** Bidders are operationalized as LLM agents, each with their own strategies and assets. Bidders actively participate in the auction, making decisions based on their current intentions, such as making a bid or withdrawing. Each action a bidder takes can sway the thoughts of other bidders, leading to a series of actions and reactions that shape the dynamics of the auction and the outcome.

Formally, given an auction with $N$ bidders and $M$ items, we use indicator $x_{i,j} = 1$ to denote bidder $i$ winning item $j$, and $x_{i,j} = 0$ otherwise. The final bid price for item $j$ is $p_j$, with its true value $v_j$. For simplicity, we assume that the true value of the item is identical for all bidders, but the bidders are **unaware** of the true value. Bidders typically aim to maximize their profits in a multi-item auction, but will be prompted to assume different objectives (see § 4.3). The utility function for such a profit-driven bidder $i$ is:

$$Maximize \quad U_i = \sum_{j=1}^{M}(v_j - p_j) \cdot x_{i,j}, \quad s.t. \quad \sum_{j=1}^{M} b_j \cdot x_{i,j} \le B_i, \quad \sum_{i=1}^{N} x_{i,j} = 1, \quad x_{i,j} \in \{0,1\}$$

where $B_i$ is the budget for the bidder $i$. Note that the bidder will not incur any budget loss from a failed bid, but may have negative profit if the winning bid exceeds its true value. Also, bidders can pursue other objectives, *e.g.*, securing a particular item or as many items as possible.

## 3.2 BIDDER AGENT ARCHITECTURE

In this section, we introduce the *bidder agent architecture* (Figure 1(B)), which utilizes the **Belief-Desire-Intention (BDI) Model** (Bratman, 1987; Georgeff et al., 1999; Andreas, 2022) as our main conceptual tool, covering the way agents are programmed to solve specific tasks, and how the agent communicates its plans and beliefs. In the case of auctions, the BDI model assigns the following attributes to agents: *1)* **Belief**: the bidder's understanding of the auction status, including the *remaining budget*, the *profits*, and *the items won* so far; *2)* **Desire (Objective)**: the bidder's objective in an auction. The most common desire is to maximize their total profit, yet some might also have non-monetary motivations, such as a personal desire to own a specific item. *3)* **Intention (Plan)**: a bidder's concrete plan to realize its desire, which may change as the auction unfolds when new information is made available.

This model offers a structured way of understanding how bidders strategize, react to new information, and adjust their plans. The outcome of each round of the auction (*e.g.*, who wins and at what price) provides new information for the bidders to update their beliefs and potentially revise their intentions for future auctions. Based on this framework, we define four actions for the bidder agent architecture: *planning*, *bidding*, *belief update* and *replanning*. In practice, we prompt LLMs with carefully designed *zero-shot* instructions (Kojima et al., 2022) without demonstrations (Brown et al., 2020; Wei et al., 2022b). The instruction prompts are showcased in Appendix A.2. A particular advantage of our LLM agents is that to elicit this information, we can directly query them by asking them textual questions at different points in the auction e.g., after each decision we can prompt them to tell us details about their understanding of their own budget, or the reasons why they made the decisions that they did, all of which helps to do belief tracking as in Richardson et al. (2022).

**Planning** Effective planning is crucial for agents to make informed decisions and well-thought-out bidding strategies that benefit both the present and future. This requires a bidding strategy in the initial step, where the agent $i$ considers its budget ($B_i$) and all the available items, and generates a **pre-auction** textual plan. This plan acts as a strategic guide for efficient resource allocation throughout the multi-item auction.

**Bidding** Plans are confirmed through execution in the bidding step. In each round, non-leading bidders who are not the highest in the previous round can either place a higher bid or withdraw, while the previous top bidder skips bidding. Initial bids start at or above the starting price, and by the final round, all but the winning bids are zero. In practice, we guide the agent to perform intermediate reasoning first before finalizing any decision.

**Belief Update**   Due to the context length limitation of LLMs, we cannot feed the entire bidding and conversation history to the agent. Therefore, we keep a dynamic memory by asking the agent to summarize the bidding history, serving as a notebook to keep track of the auction status. As shown in Figure 1(C-1), bidders' beliefs include: *1)* **Remaining Budget of Oneself**, *2)* **Total Profit of All Bidders**, and *3)* **Winning Bids of All Items**, which is generated in JSON format. The agent's belief is only updated after a bidding war for an item. By doing so, only a much shorter belief will be carried forward, preventing token overload for future items. Of course, some agents may make errors during this step, such as mistaking one's secured items or miscalculating profits. We record belief errors as an indication of LLM agents' limitations and ask the auctioneer for *belief correction* to keep the game going, akin to a bidder using a notebook and calculator in an auction.[1]

**Replanning**   An important characteristic of AUCARENA is its dynamics and ever-changing nature, making earlier agent plans prone to becoming outdated and failing to execute. Therefore, we add a replanning step for an agent to adjust its strategy based on the auction's progression and new information. After the bidding of an item, the agent reflects on its belief and previous plan and comes up with a new plan. Then, the auctioneer moves the auction forward by presenting the next item, entering another bidding, belief update, and replanning iteration.

### 3.3   SIMULATION DESIGNS IN AUCARENA

Auctions can sometimes be difficult to understand due to numerous confounding factors. Therefore, we propose three simplifying design decisions to facilitate further analysis while keeping the possibilities of more complex designs for future research, which we elaborate on in Appendix C.

**Types of Items**   We introduce two categories of items: *1)* **Cheap Items**: Valued at $2,000, with a starting price of $1,000. They are more accessible and less risky. *2)* **Expensive Items**: Valued at $10,000, with a starting price of $5,000. They are more potentially profitable if secured. This design choice allows us to study how different item values affect bidder behavior and strategies.

**Overestimation and Winner's Curse**   One intriguing aspect of auctions is the "Winner's Curse", where the winning bid exceeds an item's true value, leading to losses — a common occurrence when bidders lack precise value estimations, a typical scenario in real auctions. To replicate this, we introduce an intentional overestimation of item value. By default, we set the bidders to have an *estimated value* that is 10% higher than the *true value*. Bidders are not informed about items' true values but only estimated values. This design allows us to study the risk management skills of LLM agents, where overbids can lead to apparent victories that are, in essence, strategic losses.

**Priority Score in the Plan**   Consolidating the plans mentioned in § 3.2 is important to assess the capabilities of agents. Simplifying, a bidder's future bidding strategy can be distilled into a prioritization of remaining items. When the auction moves to the $t$-th item (item $t$), bidder $i$ assigns a three-tier priority score for each remaining item $j$ ($t \leq j \leq M$), denoted as $r_{i,j}^{(t)} \in \{1, 2, 3\}$:

**1 =** The item is of minimal importance, and consider giving it up if necessary to save money;
**2 =** The item holds value but isn't paramount, and could bid on it if you have enough budget;
**3 =** The item is of utmost importance and is a top priority.

The scoring system, which is directly generated by LLMs, provides a tangible metric to gauge a bidder's foresight and adaptability. As depicted in Figure 1(C-2,3), the agent is instructed to first engage in intermediate reasoning of the current situation before making a structured plan.

## 4   EXPERIMENTS

The experiments aim to answer three progressively advancing research questions, each targeting specific aspects of evaluation: *1)* **Rationality**: *Do agents have the basic skills for effective engagement in auctions?* (§ 4.1) *2)* **Adaptability**: *Can agents adapt themselves in a changing environment?* (§ 4.2) *3)* **Multi-agent Competition**: *How do multiple agents compete with each other under limited resources? How do they perform when compared against simple rule-based or human agents?* (§ 4.3)

---

[1]An example feedback from auctioneer is: "Bidder 1 has earned $1,000 so far. Please revise accordingly."

**Experimental Setups** We use the following SoTA LLMs as the brain of bidding agents: *1)* PaLM-2 (Anil et al., 2023) (`chat-bison-001`); *2)* Claude-Inst-1.2 (Anthropic, 2023b) (`claude-instant-1.2`), Claude-2 (Anthropic, 2023a) (`claude-2.0`); and *3)* GPT-3.5 (OpenAI, 2022) (ChatGPT, `gpt-3.5-turbo-0613`), GPT-4 (OpenAI, 2023) (`gpt-4-0613`). For every experiment, we conduct 30 simulations with varied item orders to observe average behaviors, setting the temperature at 0.7 for diversity. The typical query contains approximately 1600 tokens, with a maximum of about 4000, fitting within an LLM's 4096-token context window. The auctioneer requests a minimum 10% increase over the starting price to prevent protracted bidding, and bidders know the item order and initial prices throughout.

## 4.1 RATIONALITY OF LLM AGENT

*Can an LLM agent behave reasonably in* AUCARENA*?* We first assess the *belief tracking* and *plan following* abilities of each individual LLM, which are fundamental to forming reasonable plans and interacting with the external world. To avoid too many variables during analysis, the arena is initialized with 10 items (8 cheap ones and 2 expensive ones). Only *2 bidders* with the same setting compete in the arena, with an overestimation ratio of 10% and a budget of $10,000 or $20,000. To probe belief tracking and planning following, we ask the agents to generate structured beliefs and priority scores in the plan, as showcased in Figure 1(C).

**How accurately do agents monitor the auction environment?** To this end, we measure the failed bid rate and the belief error rate regarding the agents themselves and others. *1)* The failed bids rate is calculated in the bidding step, where we count the number of errors exhibited by LLMs in bidding, *e.g.*, bidding below the previous round's maximum or exceeding the budget. *2)* The belief error rate is computed in the belief updating step, where we count the number of errors made by different LLMs in answering their remaining budget, winning bids (for self) as well as the errors regarding the other agents' status (for others). We define the *Corrected Failure Rate*: $CFR = F/(C + F)$, where $F$ is the number of failures, and $C$ is the number of correct behaviors, which is a constant since failed behaviors are corrected. As seen in Figure 2, GPT-4 and Claude-2 excel with the lowest error rates across all three metrics, while other models frequently commit basic factual errors due to their inferior capabilities in understanding

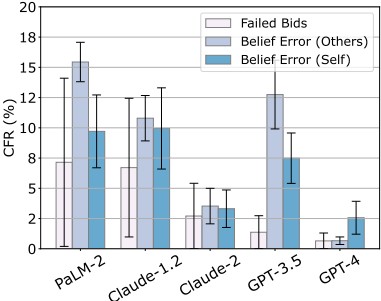

Figure 2: Corrected Failure Rate of failed bids and belief errors. The smaller the better.

context and performing elementary mathematical calculations. Note that we correct their belief errors for future bidding, so we can focus on evaluating more advanced strategic abilities.

**Do agents' actions follow their own plans?** We calculate the correlation between agent actions and the priority of each item in the agent's plan to investigate whether the agent's actions align with their plans. In general, higher-priority items motivate bidders to pursue them relentlessly, increasing their chances of winning. When bidder $i$ participates in the bidding of item $j$, in the planning, it gives the priority score $r_{i,j}$. The action is quantified as the number of engagements (bids) on this item ($n_{i,j}$), and the indication of the winning bid $x_{i,j} \in \{0, 1\}$. We then calculate the Spearman's rank correlation coefficient $\rho_s(r_{i,j}, n_{i,j})$ and $\rho_s(r_{i,j}, x_{i,j})$. The priority score could be changed during the replanning step. We use the score at two different times: the initial score $r_{i,j}^{(t=0)}$ and the current score $r_{i,j}^{(t=j)}$, where item $j$ is the next item to be bid on. In Table 1, the priority scores in the current plan strongly

| Model | Initial ($r_{i,j}^{(1)}$) | | Current ($r_{i,j}^{(j)}$) | |
|---|---|---|---|---|
| | $\rho_s(r,n)$ | $\rho_s(r,x)$ | $\rho_s(r,n)$ | $\rho_s(r,x)$ |
| PaLM-2 | 0.0574 | 0.0269 | 0.0084 | 0.0477 |
| Claude-1.2 | 0.2378 | 0.1152 | 0.4628 | 0.3700 |
| Claude-2 | **0.3110** | **0.2456** | **0.6254** | **0.5405** |
| GPT-3.5 | 0.1254 | 0.0833 | 0.4069 | 0.3124 |
| GPT-4 | 0.1570 | 0.1717 | 0.6221 | 0.3797 |

Table 1: Spearman correlation $\rho_s(\cdot)$ between the priority score of an item in bidder $i$'s initial or current plan, and the number of engagement $n_{i,j}$ on item $j$ and the winning indication $x_{i,j}$. Underlined value is statistically significant ($p < 0.001$).

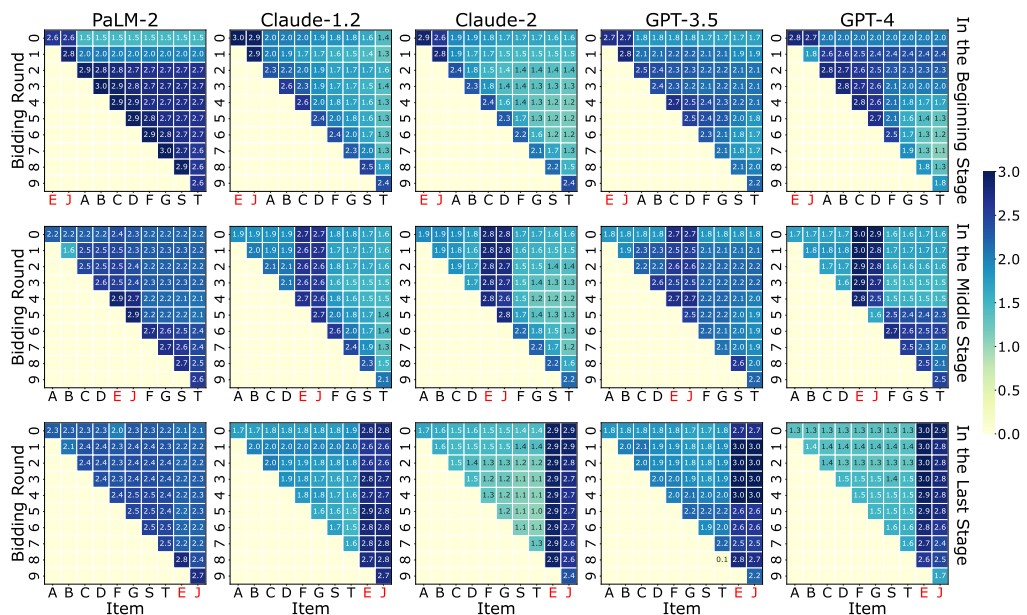

Figure 4: The heatmap of priority changes before each bidding round for adaptive, profit-driven bidders with a budget of $10,000. Item in red (E, J) denotes the expensive but more profitable items, and the others are the cheap ones. Similar results with a budget of $20,000 are in Appendix B.1.

correlate with the executions, proving the need for plan adaptation to the changing environment. Among LLM agents, Claude-2 excels in execution, with higher engagement with high-priority items (0.6254) and a better chance of winning them (0.5405). PaLM-2, however, deviates from its explicit plan and bids randomly.

## 4.2 ADAPTABILITY OF LLM AGENT

AUCARENA constantly evolves and often defies what has been planned. For example, a bidder's well-made plan for a high-priority item can be thwarted by a determined or better-funded rival bidder. In such a dynamic competitive setting, an astute agent continually adapts its strategy to ensure goals are realized in the long run. We conduct an ablation study focusing on the planning and replanning steps of the agent. We arrange an auction of 10 items, as in § 4.1.

**How does adaptability in planning influence auction outcomes?** We introduce three distinct planning strategies for bidder agents as an ablation study[2]: *1) No Planning*: the agents bid without a plan (no planning and replanning step); *2) Static Planning*: agents adhere to a fixed plan (no replanning step); and *3) Adaptive Planning*: agents adjust their plan after each bidding round. We set up an arena of 3 agents with the same LLM, but each adopts one of these three strategies, and the metric is the average total profit. Figure 3 depicts a consistent trend that having a plan uniformly benefits all and updating the plans can further improve the performance (total profit) in an auction, which is consistent with Table 1.

**Can agents strategically allocate resources according to a long-term plan?** Strategic behavior involves resource conservation for the future. To have a closer examination of plan adaptability, we present a visualization of item priority changes after each bidding round for adaptive bidders of LLMs (Figure 4). We manipulate

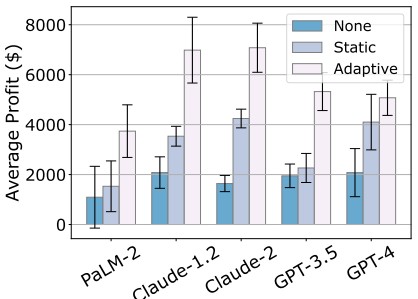

Figure 3: The average profit after competition between three agents with no, static and adaptive planning.

---

[2]While we don't explicitly ask agents to plan or replan (§ 3.2), they may still have *implicit* plans and updates.

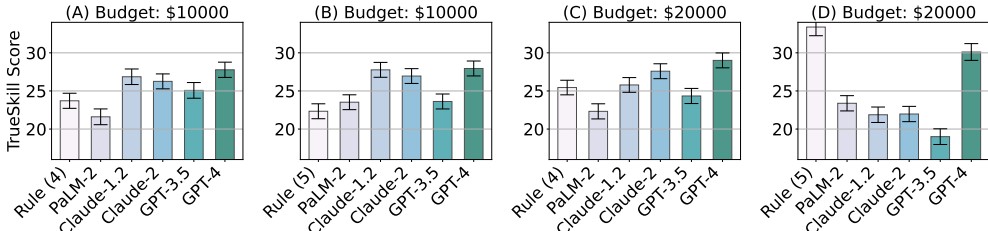

Figure 5: Intraspecific competition amongst 6 bidder agents, each motivated to maximize profit. We report the skill values of TrueSkill models $\mathcal{N}(\mu, \sigma^2)$ from two settings. Each bidder is allocated a budget of \$10,000 or \$20,000. Rule (4/5) means a rule bidder bids at most 4 or 5 times on an item.

the item order to include profitable items (expensive ones) at the start, middle, and end of the auction. Note that priorities in bidding round 0 can be viewed as the plan for static bidders, which will not be updated. We observe some interesting behaviors: *1)* Most LLMs prioritize the profitable items (E, J), regardless of their appearing stage. *2)* After bidding these items (after round 2 for the first setting and after round 5 for the second), most LLMs change the priority of the rest items, holding no conservation of budget and bidding aggressively. *3)* When E and J appear near the end, most LLMs can resist the temptation and save money for the future. *4)* PaLM-2 cannot assign reasonable priorities to items, revealing its poor strategic planning ability. We also trace the budget change when executing these plans in Appendix B.2, and has the same conclusion as the priority assignment.

### 4.3 MULTI-AGENT COMPETITION

We explore the dynamics of multi-agent competition with different LLMs as bidder agents in one arena. Analogous to the ecosystems, we delve into two types: *1)* **Intraspecific Competition**: agents share the same objective, battle to achieve individual optimization (*e.g.*, maximizing profit); *2)* **Interspecific Competition**: agents have varied objectives, which might or might not coexist together (*e.g.*, maximizing the number of acquired items and maximizing profit). For comparison, we add a *Rule Bidder* to the arena, which serves as a baseline. The rule bidder has a fixed engagement limit per item, depending on the budget, and each of their bid increases the previous highest bid, if possible, minimally (10%). We increase to 20 items, 16 cheap ones (\$2,000), and 4 expensive ones (\$10,000), so that bidders will have enough items for competition.

**Intraspecific Competition: Auction as an LLM Arena** To objectively measure and rank the performance (*i.e.*, total profit) of each bidder in such a competitive setting, we employ the TrueSkill rating (Herbrich et al., 2006; Minka et al., 2018), which estimates skill levels ($\mu$) through Bayesian statistics while considering uncertainty ($\sigma$) in their true skills. This metric reveals which agents are the most proficient in the direct competition. Figure 5 clearly illustrates the performance of the various agents in the competition, with GPT-4 standing out in particular. Rule bidders establish an intriguing baseline. With an expanded budget of \$20,000, limiting bids to a maximum of 5 appears to grant them a significant advantage over other LLMs (D); however, under a constrained \$10,000 budget (A,B), the rule-based strategy is less effective, which underscores the complexities of strategic planning within resource-limited contexts. Surprisingly, Claude-Instant-1.2 after belief correction shows superior performance over Claude-2 when operating on a \$10,000 budget (A,B).

**Interspecific Competition: Emergent Niche Specification in Auctions** Just as ecosystems host diverse species in unique niches, multi-agent environments allow agents to establish their specialized objectives. In real-world auctions, participants often have varied objectives beyond mere profit maximization. We thus simulate two types of agents: *1) Profit Bidders*, aiming for high profit, and *2) Item Bidders*, aiming for the most items won. Note that these two objectives do not yield the same results, *e.g.*, taking the most items does not necessarily mean winning more profit, and vice versa. It is conceivable that profit bidders should target costly, profitable items, while item bidders should focus on cheaper, accessible ones. To investigate this, we control the budget they have when bidding for 20 items, and assign 2 groups of bidders: 2 profit bidders and 2 item bidders, each group having a GPT-4 and a GPT-3.5 bidder. Figure 7 illustrates the density histograms and kernel density estimate plots of the winning bids from two bidder groups, from which we can observe the emergent niche specification behavior of the LLM agents. Specifically, as the budget increases from \$10,000 to

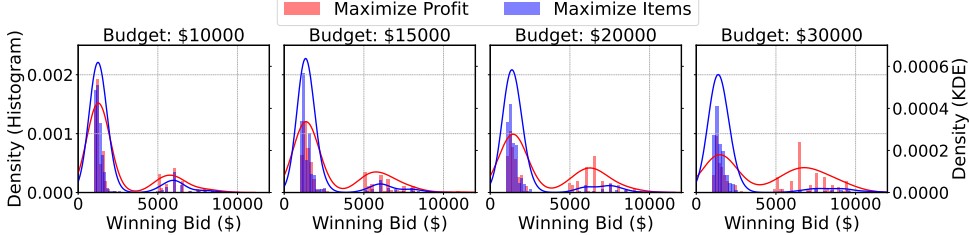

Figure 7: The interspecific competition between two groups of bidders: 2 profit bidders (red) and 2 item bidders (blue). We increase the budget of everyone and show the changes in the histograms and kernel density estimation that depict the winning bids of two bidder groups.

$30,000, the distribution gap between the two groups is enlarged. Each group starts to find its own niche, *i.e.*, growing dominance over cheap or expensive items. When the budget is limited ($10,000), profit bidders have to be more competitive on the cheap items, also leaving chances for item bidders to win more profitable items.

**Learning from Past Experiences for Human and LLM** *How do humans play in* Au-cArena*?* We recruit 5 graduate students and ask them to independently play 10 rounds against bidder agents. Following an intraspecific competition setting, we set an arena for 5 LLM bidders, 1 rule bidder (bid 5 times at most), and 1 human bidder: each has a budget of $15,000. We showcase the changes in each human bidder's ranking in 10 rounds of games in Figure 6, and find that AucArena is useful to practice their abilities. Humans generally can learn from the past and improve themselves (Banerjee et al., 2023), showing a performance boost after 2-3 rounds of games. *Can AI agents do the same?* Following existing work on learning from previous rounds of a game to improve LLM agents without parametric update (Dalvi Mishra et al., 2022; Fu et al., 2023; Zhao et al., 2023), we add a simple learning module for GPT-4, which summarizes and updates useful learnings from a brief history of the previous auction to guide

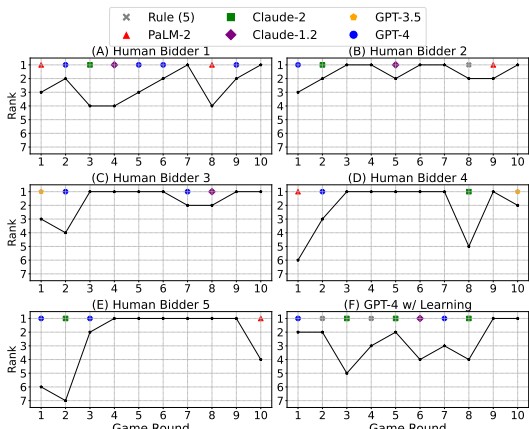

Figure 6: Ranking evolution for human bidders and a GPT-4 bidder with learning mechanism over 10 game rounds in AucArena. All of them are profit bidders.

this one, as detailed in Appendix A.2. We incorporate these learnings in the prompt, asking LLMs to use them during planning and bidding. According to Figure 6(F), GPT-4 exhibits some improvement after learning, but not obviously and consistently. More comprehensive results in Appendix B.3, and the content of learnings, show a huge gap between the learning ability of humans and LLM agents.

## 5 CONCLUSION

In this study, we propose AucArena, a multi-agent simulation in the auction setting for quantifying the strategic planning and execution abilities of LLM agents in a controllable way. We unveil a unique multi-agent simulation to scrutinize the strategic proficiencies of LLMs in auction settings. Our findings highlight the disparities in LLMs, with GPT-4 emerging as a notably superior model, and adaptive planning proving pivotal in complex multi-agent competitions. The observed ecological niche separation phenomena accentuate the interactions between agents with diverse objectives, becoming especially discernible with increased budgets. These insights underline the imperative of refining LLMs to optimize their adaptability and efficacy in varying scenarios. The emerging parallels between natural and multi-objective multi-agent competitions present intriguing future research avenues, advocating for further, innovative manipulations of our simulation method to explore these phenomena further and amplify the potential of LLMs in modeling intricate social dynamics.

## ETHICS STATEMENT

This auction simulation has the potential to allow and somehow encourage language models and humans acting as bidder agents to lie to each other, for the purpose of being strategic and victorious in an auction game. This is especially true if advanced auction rules like collaboration and negotiation between bidders are introduced into this auction arena. Agents developed in such an arena could emerge as calculative, deceptive, greedy, dishonest, and manipulative. The LLM APIs we used in this work are versions prior to September 28, 2023. The actual model checkpoints behind these APIs could be changed by OpenAI, Anthropic, and Google without our knowledge, somehow hurting reproducibility (but only to a marginal extent). However, certain GPT models with timestamps will not be secretly updated, according to OpenAI. The recruited human bidders are all compensated above the local minimum wage and consent to using their data for research purposes, and they have the authors' gratitude.

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

## A    EXAMPLES: PROMPT AND BIDDING HISTORY

### A.1    BIDDING HISTORY

An example of the bidding history is given in List 1.

Listing 1: An example auction log of a bidding war between three bidders over three items.

```
## Auction Log

### 1. Gadget B, starting at $1000.

#### 1st bid:

* Bidder 1: $1200
* Bidder 2: $1000
* None bid

#### 2nd bid:

* Bidder 1: $1300
* Bidder 2: Withdrew

#### 3rd bid:

* Bidder 3: $1400

#### 4th bid:

* Bidder 1: $1500

#### 5th bid:

* Bidder 3: Withdrew

#### Hammer price (true value):

* Bidder 1: $1500 ($2000)

### 2. Thingamajig C, starting at $1000.

#### 1st bid:
```

```
* Bidder 1: $1200
* Bidder 2: $1100
* Bidder 3: $2000

#### 2nd bid:

* Bidder 1: Withdrew
* Bidder 2: Withdrew

#### Hammer price (true value):

* Bidder 3: $2000 ($2000)

### 3. Widget A, starting at $1000.

#### 1st bid:

* Bidder 1: $1200
* Bidder 2: $1200
* Bidder 3: $1100

#### 2nd bid:

* Bidder 1: Withdrew
* Bidder 1: Withdrew

#### Hammer price (true value):

* Bidder 2: $1200 ($2000)

## Personal Report

* Bidder 1, starting with $20000, has won 1 items in this auction, with a
 total profit of $500.:
  * Won Gadget B at $1500 over $1000, with a true value of $2000.

* Bidder 2, starting with $20000, has won 1 items in this auction, with a
 total profit of $800.:
  * Won Widget A at $1200 over $1000, with a true value of $2000.

* Bidder 3, starting with $0, has won 1 items in this auction, with a
total profit of $0.:
  * Won Thingamajig C at $2000 over $1000, with a true value of $2000.
```

A.2   INSTRUCTION PROMPTS

A complete bidder agent has these functions: planning, bidding, belief update, replanning (§ 3.2), and learning (§ 4.3). Here, we report the arguments of each function as follows:

- Planning: System Message, Initial Belief, Planning Instruction;
- Bidding: System Message, (Re-)Plan Instruction, (Updated) Plan, Bidding History, Bid Instruction;
- Belief Update: System Message, Bidding History, Belief Update Instruction;
- Replanning: System Message, Updated Belief, Replanning Instruction;
- Learning: System Message, Auction Log, Previous Learnings, Learning Instruction.

The System Message and Planning, Bidding, Belief Update, Replanning, and Learning instructions are shown in Listing 2. We keep the instructions as general as possible, providing only the necessary rules of the auction and examples of output format for parsing. We try to not provide any examples

of concrete auction strategies for in-context learning to avoid any unintended biases. Due to budget limits, we do not have the resources to rigorously evaluate more forms of instruction designs like in the experiments, other than some prompt engineering endeavors during the development period.

Listing 2: The System Message and Planning, Bidding, Belief Update, Replanning, and Learning Instructions.

```
System Message:

You are {name}, who is attending an ascending-bid auction as a bidder.
This auction will have some other bidders to compete with you in bidding
wars. The price is gradually raised, bidders drop out until finally only
one bidder remains, and that bidder wins the item at this final price.
Remember: Your primary objective is to secure the highest profit at the
end of this auction, compared to all other bidders.

Here are some must-know rules for this auction:

1. Item Values: The true value of an item means its resale value in the
broader market, which you don't know. You will have a personal estimation
 of the item value. However, note that your estimated value could deviate
 from the true value, due to your potential overestimation or
underestimation of this item.
2. Winning Bid: The highest bid wins the item. Your profit from winning
an item is determined by the difference between the item's true value and
 your winning bid. You should try to win an item at a bid as minimal as
possible to save your budget.

Planning Instruction:

As {bidder_name}, you have a total budget of ${budget}. This auction has
a total of {item_num} items to be sequentially presented, they are:
{items_info}

-----------------------------------------------------------------------

Please plan for your bidding strategy for the auction based on the
information{learning_statement}. A well-thought-out plan positions you
advantageously against competitors, allowing you to allocate resources
effectively. With a clear strategy, you can make decisions rapidly and
confidently, especially under the pressure of the auction environment.
Remember: Your primary objective is to secure the highest profit at the
end of this auction, compared to all other bidders.

After articulate your thinking, in you plan, assign a priority level to
each item. Present the priorities for all items in a JSON format, each
item should be represented as a key-value pair, where the key is the item
 name and the value is its priority on the scale from 1-3. An example
output is: {{"Fixture Y": 3, "Module B": 2, "Product G": 2}}. The
descriptions of the priority scale of items are as follows.
    * 1 - This item is the least important. Consider giving it up if
    necessary to save money for the rest of the auction.
    * 2 - This item holds value but isn't a top priority for the bidder.
    Could bid on it if you have enough budget.
    * 3 - This item is of utmost importance and is a top priority for the
     bidder in the rest of the auction.

Bidding Instruction:

Now, the auctioneer says: "Attention, bidders! {num_remaining_items} item
(s) left, they are: {item_info}. Now, please bid on {cur_item}. The
starting price for bidding for {cur_item} is ${cur_item_price}. Anyone
interested in this item?" / "Thank you! This is the {bid_round} round of
bidding for this item: {bidding_history}. Now we have ${highest_bid} from
```

```
  {highest_bidder} for {cur_item}. The minimum increase over this highest
bid is ${min_increse}. Do I have any advance on ${highest_bid}?"

------------------------------------------------------------------------

As {bidder_name}, you have to decide whether to bid on this item or
withdraw and explain why, according to your plan{learning_statement}.
Remember, Your primary objective is to secure the highest profit at the
end of this auction, compared to all other bidders.

Here are some common practices of bidding:
1. Showing your interest by bidding with or slightly above the starting
price of this item, then gradually increase your bid.
2. Think step by step of the pros and cons and the consequences of your
action (e.g., remaining budget in future bidding) in order to achieve
your primary objective.

Give your reasons first, then make your final decision clearly. You
should either withdraw (saying "I'm out!") or make a higher bid for this
item (saying "I bid $xxx!").
```

**Belief Update Instruction**:

```
Here is the history of the bidding war of {cur_item}:
"{bidding_history}"

The auctioneer concludes: "{hammer_msg}"

------------------------------------------------------------------------

Congratulations! You have won {item} at {bid_price} / You have lost {item
}.
As {bidder_name}, you have to update the status of the auction based on
this round of bidding. Here's your previous status:
```
{prev_status}
```

Summarize the notable behaviors of all bidders in this round of bidding
for future reference. Then, update the status JSON regarding the
following information:
- 'remaining_budget': The remaining budget of you, expressed as a
numerical value.
- 'total_profits': The total profits achieved so far for each bidder,
where a numerical value following a bidder's name. No equation is needed,
 just the numerical value.
- 'winning_bids': The winning bids for every item won by each bidder,
listed as key-value pairs, for example, {{"bidder_name": {{"item_name_1":
 winning_bid}}, {{"item_name_2": winning_bid}}, ...}}. If a bidder hasn't
 won any item, then the value for this bidder should be an empty
dictionary {{}}.
- Only include the bidders mentioned in the given text. If a bidder is
not mentioned (e.g. Bidder 4 in the following example), then do not
include it in the JSON object.

After summarizing the bidding history, you must output the current status
 in a parsible JSON format. An example output looks like:
```
{{"remaining_budget": 8000, "total_profits": {{"Bidder 1": 1300, "Bidder
2": 1800, "Bidder 3": 0}}, "winning_bids": {{"Bidder 1": {{"Item 2":
1200, "Item 3": 1000}}, "Bidder 2": {{"Item 1": 2000}}, "Bidder 3":
{{}}}}}}
```
```

**Replanning Instruction**:

```
The current status of you and other bidders is as follows:
```
{status_quo}
```

Here are the remaining items in the rest of the auction:
"{remaining_items_info}"

As {bidder_name}, considering the current status{learning_statement},
review your strategies. Adjust your plans based on the outcomes and new
information to achieve your primary objective. This iterative process
ensures that your approach remains relevant and effective. Please do the
following:
1. Always remember: Your primary objective is to secure the highest
profit at the end of this auction, compared to all other bidders.
2. Determine and explain if there's a need to update the priority list of
 remaining items based on the current status.
3. Present the updated priorities in a JSON format, each item should be
represented as a key-value pair, where the key is the item name and the
value is its priority on the scale from 1-3. An example output is: {{"
Fixture Y": 3, "Module B": 2, "Product G": 2}}. The descriptions of the
priority scale of items are as follows.
    * 1 - This item is the least important. Consider giving it up if
    necessary to save money for the rest of the auction.
    * 2 - This item holds value but isn't a top priority for the bidder.
    Could bid on it if you have enough budget.
    * 3 - This item is of utmost importance and is a top priority for the
     bidder in the rest of the auction.
```

**Learning Instruction**:

```
Review and reflect on the historical data provided from a past auction.

{past_auction_log}

Here are your past learnings:

{past_learnings}

Based on the auction log, formulate or update your learning points that
could be advantageous to your strategies in the future. Your learnings
should be strategic, and of universal relevance and practical use for
future auctions. Consolidate your learnings into a concise numbered list
of sentences.
```

## B  ADDITIONAL RESULTS

### B.1  ADAPTABILITY VISUALIZATION FOR AGENTS WITH A BUDGET OF $20,000

Enriching § 4.2, as shown in Figure 8, we visualize the priority changes in the plan for all models when the budget is $20,000. The results echo those of Figure 4 (budget: $10,000), except PaLM-2, most LLMs demonstrate reasonable strategic planning capabilities and can conserve budgets for the future. Note a slight difference from Figure 4 that a bidder with a budget of $10,000 typically has a lower priority for Item J in bidding round 1 than that of $20,000. This is because both Item E and Item J are worth $5,000, and a bidder won't have enough budget left to bid on Item J if it has already won Item E.

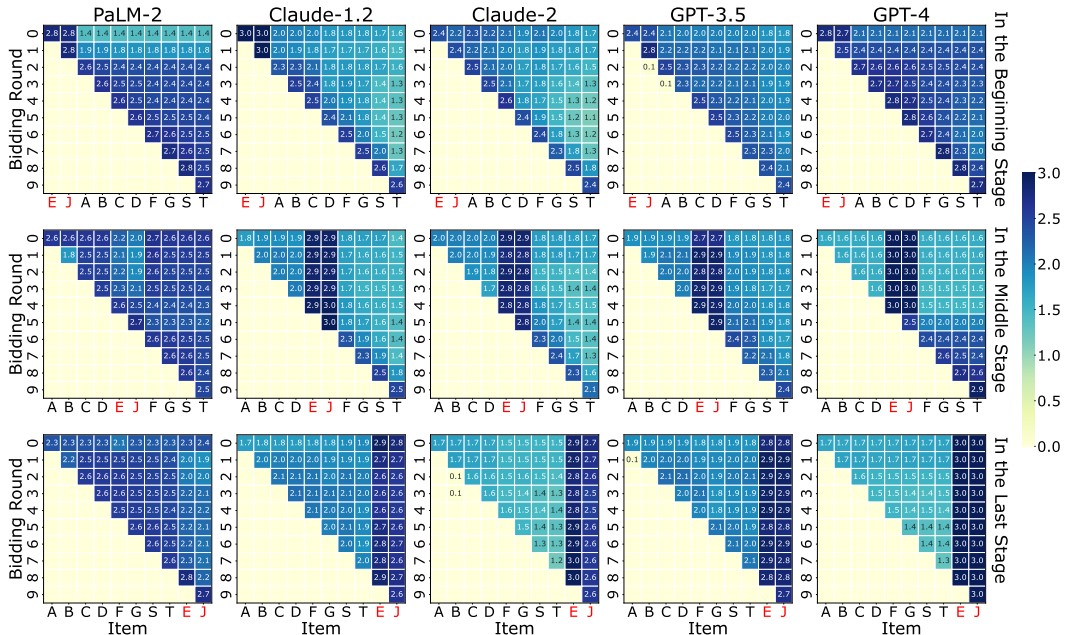

Figure 8: The heatmap of priority changes after each replanning step for the adaptive bidder of different LLMs when the budget is $20,000. Item* (E, J) denotes the expensive items, and the others are the cheap ones.

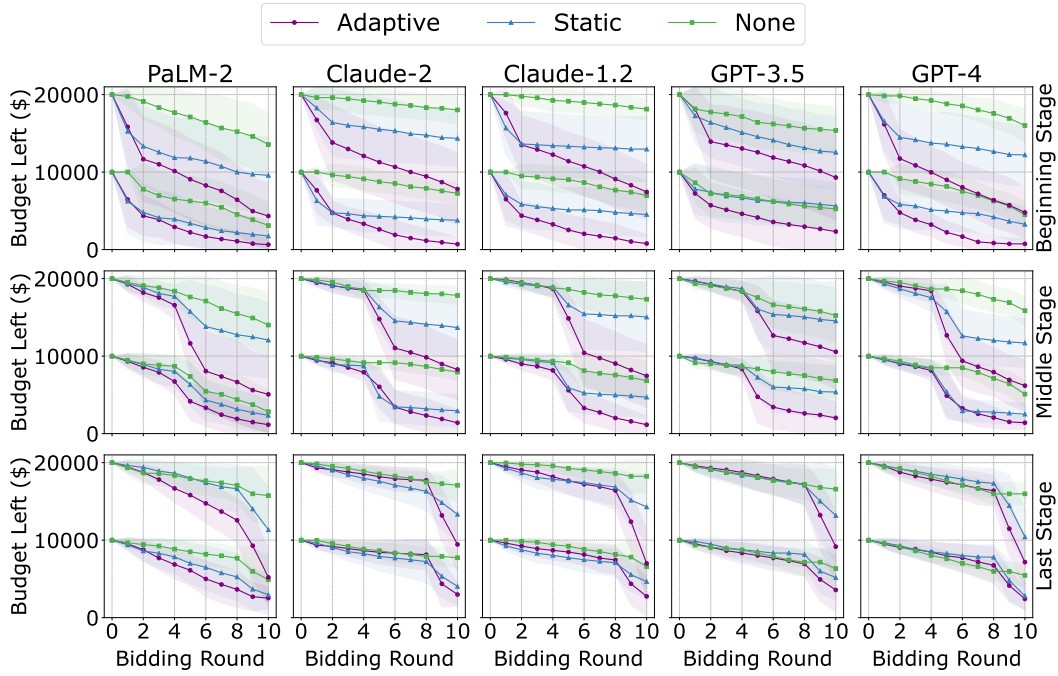

Figure 9: The changes of the remaining budget as the auction moves on. The item order is different in each row, where two expensive items (§ 3.3) appear in different stages of the auction: in the beginning (1-2), middle (5-6) and last (9-10) stage.

### B.2 TRACING REMAINING BUDGET

Other than the changing priority scores as shown in the plan visualization (Figure 4 and Figure 8), we would also like to see how these changes are carried out in practice. Inheriting the previous three-bidder setting, we study two budget settings ($10,000 or $20,000) of these LLM agents. Note that they always have knowledge of the entire item schedule. In Figure 9, we record the change in budget as the auction goes on, which reflects the details of how they execute and adapt their strategies. We find that both adaptive and static bidders can save money to bid on expensive items, whereas bidders with no planning seem too conservative. However, static bidders do not bid much on future items when expensive items have been auctioned in the beginning or middle stage due to their static prioritization. In contrast, adaptive bidders generally make better use of their budget, spending the most of it in the auction. They also allocate their budget well, saving for the expensive items but also actively bidding for future items, as evidenced by their priority changes. As for differentiating LLM agents, LLMs exhibit a different level of strategic ability in resource allocation, which helps us understand their strategic abilities, *e.g.*, PaLM-2 spends its budget quickly before the profitable items are presented in a later stage, not showing a clear money-saving behavior pattern compared with other models.

### B.3 ADDITIONAL RESULTS FOR HUMAN AND LLM BIDDERS WITH LEARNING

**Additional Results for LLM Agents with Learning** Enriching § 4.3, as shown in Figure 10, we display the progression of ranks for all other models with the learning mechanism over 10 rounds of AUCARENA. The results reveal that, worse than GPT-4, other LLMs fail to show any significant improvement after learning, highlighting a considerable gap between human and LLM learning abilities.

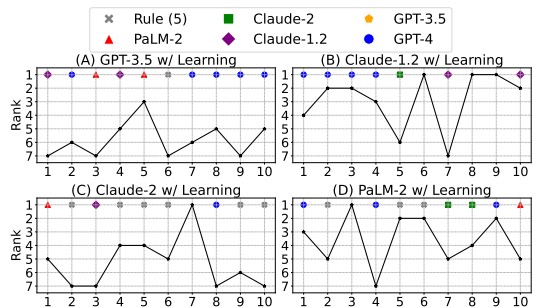

Figure 10: The progression of ranks for different LLM bidders with learning mechanism, respectively, over 10 rounds of AUCARENA. All of them are profit bidders.

**Takeaways from Bidders After 10 Rounds** In Listing 3, we report the takeaways and learnings after 10 rounds of auction games in AUCARENA for both five human bidders and GPT-4 with the learning module. Note that some human bidders find some practical strategies in auctions, such as pushing the price to 150% of the starting price of every item to drain others' budgets so that one can aim for later items. These examples indicate that there is still work to do for LLM agents to effectively do continual learning.

Listing 3: Strategic learnings and takeaways from human bidders and LLM-agents after 10 rounds of games in auction arena.

```
Human Bidder #1:
All in all, every bidder has specific strategies or tendencies.
bidder1: Tend to buy nothing
bidder2: Waiting for opportunities, likes to pick up leaks.
bidder3: Most of the time, it is conservative and likes to withdraw, and
rarely gives a price exceeding 1.6 times the starting price.
bidder4: Pay more attention to large items of $5000 because it is
possible to obtain higher profits. It's easy to withdraw from the initial
 1000 small items. If the bidding price exceeds 1.6 times, it is easy to
withdraw the bidding.
bidder5: Sometimes it makes sudden decisions and buys goods at irrational
 prices (buy a $1000 item by more than $10000).
bidder6: Love buying from the beginning, no matter how much the item
costs, until the money runs out.
bidder7: Pay attention to large items of $5000, pick up leaks if the
price of $1000-items are below $1600. I bid up the price before bidder6
finished buying. Try to let bidder2&bidder4 get $5000-items with a price
```

higher than $8000 (so the left money is lowwer than $7000 and I can pick
the ups of other 5000-items). Generally speaking, 1.5 times the starting
price is a dividing line for me to measure whether a product is a good
deal.
I think the number of $5000-items and the order in which they appear have
 a great impact on the strategy of the game. If $5000-items appear later,
 the previous $1000-items will be easily missed by these models so that I
 can pick the leaks. If there are more $5000-items, then the variables in
 the game will become larger and it will be harder for me to control the
behavior of other models.

**Human Bidder** #2:
1. Primarily focus on the four 5k items, keeping at least 8k on hand
until at least the penultimate 5k item, to prevent any bidder from buying
 the 5k item at a low price and making a substantial profit. Each 5k item
 should be bid up to at least 7k.
2. For the 1k items: Since bidder 6 will continuously increase the price
in the auction of the 1k items for profit, it's necessary to compete with
 them, raising the price to 1700. (Since bidder 6 will also buy 5k items,
 it's crucial to ensure they don't make too much on the 1k items.)
3. The last few 1k items may be priced over 2k, so don't expect to make a
 profit from them.

**Human Bidder** #3:
It is evident that bidder 5 and bidder 6 exhibit a propensity for
purchasing items priced at 2000 units, whereas bidder 3 and bidder 2
demonstrate a preference for items priced at 5000 units. Building upon
these observations, we can formulate a strategic approach:
Purchase a total of 6 items: 5 priced at 2000 units (with a target price
below 1500) and 1 priced at 10000 units (with a target price below 7500).
1. Early Game: Implement a proactive bidding strategy by offering higher
prices, such as bidding 3000 for 2000-priced items and 11000 for 10000-
priced items. This approach aims to deplete the funds of other players
and reduce the likelihood of intense competition in the mid to late game.
 The primary focus here is on bidders 5 and 6, as bidder 6 frequently
engages early in the game.
2. Mid-Game: Concentrate on acquiring 2000-priced items for 1500 units or
 less and securing 10000-priced items for 7500 units or less. Initiate
bidding around 6000 units for 10000-priced items, and if the price
exceeds 7500 units, consider abandoning the item. The objective is to
encourage other players to purchase items at prices exceeding 7500 units,
 thereby preventing them from acquiring two 10000-priced items
consecutively.
3. Exercise caution when making purchases near the end of the game.
Players with remaining funds often engage in continuous bidding, leading
to higher prices, particularly for items priced at 2000 units, which may
exceed 1800 units.

**Human Bidder** #4:
In earlier experiments, it was observed that bidders who could start with
 a bid of 5000 in the later stages often ended up winning. This is
because, by the latter stages, many bidders had already spent a
significant amount of money and did not have enough left to bid on items
with a starting price of 5000. Additionally, given that the order of the
items is random, there's a chance that an item priced at 5000 could
appear twice, both in the first ten rounds and the last ten rounds of a
20-round auction. Thus, in subsequent tests, the strategy was to drive up
 the prices in the first ten rounds to deplete the machine bidders' funds
. Specifically, human bidder would start by bidding 1.5 times the
original price to drive up the costs and then opt out of the following
bids. In the last ten rounds, human bidder would engage in bidding,
always increasing bid by the minimum increment over the highest previous
bid. Using this strategy, human bidder often managed to win items with a

starting price of 5000 for a bid of 7500 or less, as well as several
items with a starting price of 1000 for 1500 or less.

**Human Bidder** #5:
I find that language models tend to spend all the money in the first few
rounds, and the later bids are much less than the previous ones. From a
profit perspective, items worth 2000 generally bring a relatively low
profit, so we can spend money on items worth 10000. For items worth 2000,
 each bid is around 1700, preventing others from getting their money at a
 low price and consuming their money. For items worth 10000, i bid around
 8000 in the first two times. I find other auctioneers usually follow me,
 which results in others earning less. Afterwards, i can bid for a low
price (6500/7000). Following this strategy can usually win!

GPT-4 **Bidder**:
1. Patience and endurance often pay off, as many winning bids emerge from
 consistent and steady increases rather than an initial high bid.
2. Understanding the true value of an item is key to determining your
maximum bid and avoiding overpayment.
3. Regular participation in the bidding process is crucial. Winners often
 demonstrate active engagement throughout the auction.
4. Be observant of other bidders' behaviors. Early withdrawals can
present opportunities for a winning bid.
5. Avoid 'auction fever,' or the excitement that can lead to overbidding.
 Stick to your predetermined maximum bid.
6. A significant jump in your bid can deter other bidders but should be
within your budget and strategically considered.
7. Don't be discouraged if you're not leading early in the auction. Many
winners aren't the early frontrunners.
8. Observing the bidding behavior of others can provide valuable insights
 to inform your strategy.
9. Be flexible and adaptable in your bidding strategy, ready to revise it
 based on the actions of other bidders.
10. Maintain financial discipline. Be aware of your budget and do not
exceed it.
11. Begin active bidding when the price is within a reasonable range of
your valuation. Early active bidding might escalate the prices quickly.
12. Sometimes, holding back and not rushing into bidding can provide
valuable insights into other bidders' behaviors.
13. Regularly reassess your strategy during the auction based on the
behavior of other bidders and the current highest bid.
14. Winning at an auction does not always mean paying the highest price;
sometimes, it's about outlasting the competition and having the last
valid bid when others have withdrawn.
15. It's important to not get emotionally attached to an item to avoid
bidding beyond its worth or your budget.
16. A successful bidder knows when to fight for an item and when to let
it go, understanding that not every item is worth winning at any cost.
17. Understanding the dynamics of an auction and the psychology of other
bidders can be more valuable than the financial capacity to bid high.
18. Always have a well-thought-out exit strategy; knowing when to stop is
 as important as knowing when to bid.

GPT-3.5 **Bidder**:
1. Setting a reserve price can help ensure that the starting bid is
closer to the true value of the item and discourage low initial bids and
subsequent withdrawals.
2. Analyzing bidding patterns and strategically increasing bids in small
increments can help maintain a competitive advantage.
3. Considering the motivations and strategies of other participants and
adjusting bidding strategies accordingly can improve the chances of
success.
4. Continuously monitoring the number of active bidders and adjusting
strategies as the auction progresses, taking into account decreasing

bidding activity and the potential for remaining bidders to withdraw, is
important.
5. Evaluating the perceived value of an item and considering it alongside
 the final hammer price is crucial, as the final price can be influenced
by bidding strategies and competitive dynamics among bidders.
6. Anticipating potential bidding dynamics and withdrawals based on
outbidding by previous bidders and adjusting strategies accordingly can
improve competitiveness.
7. Carefully considering the starting price of an item and its impact on
bidder engagement and competition is important, as it can significantly
affect bidding activity and the final price.
8. Regularly assessing and adapting bidding strategies based on the
behavior and patterns of participants can help inform future decision-
making.
9. Implementing bidding increments that encourage competitive bidding
while avoiding excessive jumps that can discourage participation is
beneficial.
10. Using historical auction data to analyze and understand bidding
behavior, identifying trends and patterns, can inform future bidding
strategies.

Claude-Instant-1.2 **Bidder**:
1. Monitor the number of active bidders on each item to assess
competition level and determine an appropriate maximum budget that allows
 winning bids without overpaying.
2. Strategically withdraw counterbidding when only one opponent remains
and the price exceeds your maximum budget, rather than unnecessarily
increasing bids just to determine the highest bidder.
3. Successively outbidding opponents by small incremental amounts is an
effective tactic to win items when within budget, avoiding unnecessarily
driving prices much higher through large counterbids against multiple
bidders.

Claude-2 **Bidder**:
1. Identify and track competitors who habitually overbid to exploit their
 tendencies.
2. Make bold opening bids to test waters, but be ready to withdraw if
competition emerges.
3. Bid just enough to stay competitive when facing multiple engaged
bidders.
4. Restart stalled bidding below true value if possible to extract better
 deals.
5. Time bids for maximal impact when competitors seem distracted or
disengaged.
6. Withdraw quickly at preset limits to avoid overpaying in the heat of
bidding.
7. Cautiously bid on items with little early engagement to get good value
.
8. Be willing to incrementally increase bids on coveted items up to your
limit.
9. Use small bid bumps to win uncontested items without overpaying.
10. Balance bid amounts based on number of competitors and intensity of
engagement.

PaLM-2 **Bidder**:
1. Bidders often withdraw their bids, so it is important to be prepared
to increase your bid if you are interested in an item.
2. The number of bidders and the level of competition can vary
significantly from auction to auction, so it is important to be prepared
to adjust your bidding strategy accordingly.
3. The starting price of an item is not necessarily indicative of its
final sale price.

```
4. Some bidders may be more willing to pay a higher price for an item
than others, so it is important to be aware of the competition and be
prepared to adjust your bid accordingly.
5. The true value of an item may not be known until the end of the
auction, so it is important to carefully consider your maximum bid before
 placing it.
6. It is important to have a clear bidding strategy in place before the
auction starts, and to stick to it.
```

## C   MANIPULATING AUCARENA IN THE FUTURE

Limited by a research paper, we keep AUCARENA to be simple and controllable for a better under-standing and evaluation of LLM agents. For example, we design limited types of item values, the same budget for everyone, only two objectives for bidders, and no autonomy for the auctioneer. We believe there are countless combinations of settings that this simulated arena can offer, which we listed as follows to invite future research:

- Diverse and Personalized Values for Items: Items can have various estimated and actual values for different bidders. Some of them may even be emotional values. For example, a bidder who is into modern art could have higher estimated values for such an item.

- Different Desires for Bidders: Each bidder could pursue a different agenda, such as securing certain items, maximizing profit, or the number of items. Some of these desires may even be malicious, such as deliberately elevating the bidding war so that others will have to pay more for an item, even if the bidder does not want this item. It would be interesting for an agent to tell whether the opponent in the bidding war is a shill bidder or not, which tests the Theory-of-Mind abilities (Sclar et al., 2023; Shapira et al., 2023) of autonomous agents.

- Negotiation Between Bidders: Bidders could negotiate with each other for cooperation so that the bidding wars might not be so intense and they can have their arranged items at a much lower price. This will test the ability to negotiate, collaborate, and even know how and when to lie, which are all rather important in strategic tasks (, FAIR).

- Seller Agents: To improve the complexity and make the simulation more realistic, the auctioneer or the seller could also have its own agenda, such as selling the items at a higher price as possible. Then, the auction would be even more tricky since bidders face the competition of both fellow bidders and the auctioneer. How to find an optimal trajectory in such an environment would be a rather interesting question.

- Life-long Learning for LLM Agents: It is possible for an LLM agent to learn from previous experience, as pointed out by many existing works. How to let an LLM agent effectively and continually learn from dozens or even hundreds of rounds of simulations would be interesting and challenging. Also, an even more interesting question would be: can LLM agents discover some novel bidding strategies that humans did not think of by running such simulations multiple times?

