# OpenReview forum: "Put Your Money Where Your Mouth Is: Evaluating Strategic Planning and Execution of LLM Agents in an Auction Arena"
_ICLR.cc/2024/Conference — ICLR 2024 Conference Withdrawn Submission_

### Official Review · Reviewer_Yv5K · 2023-10-24

**Soundness:** 3 good
**Presentation:** 3 good
**Contribution:** 3 good
**Rating:** 6
**Confidence:** 4

**Summary:**

This paper present AucArena, an open ascending auction protocol for LLM-based bidding agent. The auction protocol specifies the value of items and communication process. The LLM agent is assumed to follow a certain architecture, which consist of (1) building belief, e.g., the sufficient statistics like remaining budget, profits (2) Desire, the objective that a bidder tries to optimize and (3) planning, the bidder's strategy to fulfill its goal. These components structuralized an LLM agent's decision-making process, and is purely generated by LLM itself. The evaluation studies the correlations between the agents' actual bidding behaviors, and the output of each of these components.

**Strengths:**

The presentation of the paper is good. The usage of LLMs as bidding agents are novel and promising.

**Weaknesses:**

Some of the settings and experimental results are not clear. Please see the question section.

**Questions:**

I like the setting of using auctions as benchmark to evaluate LLMs. However, there are a few questions I have in mind:

1. About the auction protocol. Section 3.1 wrote the true value of a particular type of item is the same for all players, but the players cannot observe. This is very confusing to me. In a standard auction setting a player will have a private valuation, or some prior distribution of the ground truth value. The distributions could be correlated among different players, but I cannot understand the setup of not having any information of the ground truth values at all. Also, can the bidder observe its own utility every round? If it does then it can infer the ground-truth utility right? Can the author please clarify what is the observation space for a bidder each round.

2. In section 3.3. Also I believe winner's curse happens when the players' beliefs of valuations are positively correlated.

3. In the adaptivity experiment, what is the learning rule mainly about? And why not incorporate this learning rule in previous experiments?

---

> ### Author Response · Authors · 2023-11-18
>
> Thank you for your encouraging feedback! Here are the answers to your questions:
>
> > Q1.1: I cannot understand the setup of not having any information of the ground truth values at all. Also, can the bidder observe its own utility every round? If it does then it can infer the ground-truth utility right? Can the author please clarify what is the observation space for a bidder each round.
>
> A1.1: We would like to clarify that the bidders have an estimation of the true value. This is because in the traditional auction scenario, the bidders usually have an estimated value for the item based on their prior knowledge, market research or historical prices. However, this estimated value may not exactly be the item's true value [1], due to overestimation or underestimation of the item value (in our case for example, by a percentage of 10%). We follow this setting to approach real-world cases, so that we can study the risk management skills of LLM agents (e.g., winners' curse), where overbids can lead to apparent victories that are actually strategic losses. We will clarify this detail in the revision.
>
> > Q1.2: Can the bidder observe its own utility every round? If it does then it can infer the ground-truth utility right? Can the author please clarify what is the observation space for a bidder each round.
>
> A1.2: Yes, each bidder can observe its own utility (i.e., profit or items won) every round to infer the ground-truth utility. For each round, the observation space for each bidder includes: the remaining budget of oneself (budget information is not shared), profits so far for everyone (can be calculated based on the announced true value and the winning bid of an item), the items won so far and their winners, and the auction bidding history.
>
>
> > Q2: In section 3.3. Also I believe winner's curse happens when the players' beliefs of valuations are positively correlated.
>
> A2: Yes, it seems players' beliefs of high valuations are more likely to lead to the winner's curse. However, the winner's curse doesn't necessarily happen if bidders are risk-aware and consider certain auction strategies. For example, they might take this risk into consideration to anticipate the winner's curse, and intentionally undervalue an item or limit their maximum bid (e.g., hedging strategy) to avoid paying too much for it.
>
>
> > Q3: In the adaptivity experiment, what is the learning rule mainly about? And why not incorporate this learning rule in previous experiments?
>
> A3: Just to clarify, did you mean the learning module from Sec 4.3 (Figure 6)? This learning module enables agents to learn from the logs of previous auctions, and summarize some textual takeaways to guide future auctions. We didn't incorporate this since it does not bring much improvement to the agent in general, and it requires previous context for the agent to learn from, whereas most of the experiments are conducted in an "zero-shot" manner.
>
> There are some works on non-parametric learning (i.e., no training for the model) for language agent. The results somewhat contradict those found in [2], where a similar non-parametric learning technique was applied to tasks like HotpotQA. We think this could be because the insights gained from such NLP tasks, such as "don't neglect a negation" or "decompose a sentence" are rather static. However, they might not apply to our auction arena. Unlike the static tasks, our auction environment is dynamic and unpredictable. The past experiences may not be effective in a new context influenced by fresh rounds of conversations and bids.
>
>
> [1] Klemperer P. Auction theory: A guide to the literature[J]. Journal of economic surveys, 1999, 13(3): 227-286.
>
> [2] Zhao, Andrew, et al. "Expel: Llm agents are experiential learners." arXiv preprint arXiv:2308.10144 (2023).

---

### Official Review · Reviewer_TdM7 · 2023-10-29

**Soundness:** 2 fair
**Presentation:** 2 fair
**Contribution:** 2 fair
**Rating:** 3
**Confidence:** 4

**Summary:**

This paper introduces AucArena, a simulated auction environment for language model agents, and conducts comprehensive benchmarks among PaLM-2, Claude-1.2, Claude-2, GPT-3.5, and GPT-4.
The authors test language model agents based on Belief-Desire-Intention (BDI) Model framework, where the agents are expected to keep the state of auctions (remaining budget, total profit, winning bids of all items) in mind, to follow the given instructions, and to replan the future bids dynamically. While language model agents can follow the given instructions during the auction (budget, priorities, objective, etc.), human players or even rule-based agents still work as competitive baselines.

**Strengths:**

### quality and clarity
- This paper is clearly written and easy to follow.

 ### originality and significance
- The auction might be a unique setting for language model agents.
- In contrast to Reflexion [1],  tested on ALFWorld (a housing simulator) and HotpotQA (Wikipedia-based QA), the self-improvement loop from the previous experience is not observed (Section 3.3), which might be an interesting finding.

[1] https://arxiv.org/abs/2303.11366

**Weaknesses:**

- While auction might be a unique setting, it is unclear what aspects of language model agents would be evaluated through this benchmark.
- (Re)Planning [1,2], Instruction-following [3], and belief (internal state of the agents) [4,5] is a typical design for language model agents. It does not seem as a novel proposal or evaluation of language model agents.
- The authors mentioned AgentBench in Section 2 as `... are limited in terms of evaluating LLMs in dynamic environments.`, but in fact, they have static offline task planing evaluation (Mind2Web).
- I'm not sure if the variety of items is enough (\\$2000 and \\$10000). Some justification would be helpful.
- The descriptions of AucArena in Section 3.1 `For simplicity, we
do not introduce additional values such as various personal preferences and emotional value of items.` and the descriptions of BDI model in Section 3.2 `... yet some might
also have non-monetary motivations, such as a personal desire to own a specific item.` seem contradictory.


[1] https://arxiv.org/abs/2303.17491

[2] https://arxiv.org/abs/2305.16653

[3] https://arxiv.org/abs/2207.01206

[4] https://arxiv.org/abs/2210.03629

[5] https://arxiv.org/abs/2207.05608

**Questions:**

- How many participants do you have in ActArena? One LLM-based agent you are focusing on, and others are rule-based agents?
- While GPT-4 achieves the lowest Corrected Failure Rate in Figure 2, the average profit of Claude-1.2/2 with adaptive planning is higher than GPT-4 in Figure 3 (GPT-3.5 does not seem a bad bidder here). Why does this happen?

---

> ### Author Response · Authors · 2023-11-18
> **Response #1**
>
> Thank you for the valuable comments! Please find our responses to your questions and concerns below.
>
> > Q1: While auction might be a unique setting, it is unclear what aspects of language model agents would be evaluated through this benchmark.
>
> To emphsize again, our main focus was measuring the strategic planning and execution capabilities of LLM agents in dynamic scenarios, for which the auction setting is well suited. We conducted a detailed breakdown and assessment. A reminder of these different aspects:
>
> - In Sec 4.1, Rationality represents the ability of strategic planning, specifically divided into whether the agent can correctly perceive the environment (measured by CFR) and whether the agent can implement the plan into the correct actions (measured by the correlation between the priority score for a specific item and the number of bids).
> - Sec 4.2's Adaptability is the measure of the agent's execution ability in a "dynamic" environment, specifically including the impact of Adaptive Planning and whether resources can be allocated correctly. Additionally, we conducted a competitive analysis in a multi-agent environment, comparing the strengths and weaknesses of different LLMs.
>
>
> > Q2: (Re)Planning [1,2], Instruction-following [3], and belief (internal state of the agents) [4,5] is a typical design for language model agents. It does not seem as a novel proposal or evaluation of language model agents.
>
> We acknowledge that the fundamental abilities of language agents, such as (re)planning, instruction-following, and belief tracking, are indeed well-established in the literature. However, the primary contribution of our work is not in the exhibit of these basic abilities for language agents per se, but rather about how they are applied strategically and adaptively in a real-time, dynamic setting.
>
> Our research introduces a dynamic auction arena that uniquely challenges these agents in strategic decision-making, adaptability, and execution under competitive and resource-constrained conditions. This environment significantly differs from traditional static benchmarks and provides a more realistic and unpredictable setting, where the strategic application of these foundational abilities is critical.
>
> Moreover, the evaluation of agents in this auction arena offers valuable insights into their strategic capabilities and limitations. It demonstrates how these agents perform in real-time, adapt to rapidly changing scenarios, and make strategic decisions to achieve their objectives. This aspect of our research is particularly relevant, as it sheds light on the current state of agent abilities and highlights areas for future development in agent design to enhance their strategic planning and execution.
>
> In summary, the novelty of our work lies in the application of established agent abilities in a unique, competitive auction setting, providing new insights into the strategic capabilities and limitations of current language model agents. Our research contributes significantly to the field by moving beyond static evaluations and demonstrating the potential and challenges of applying these agents in dynamic, real-world scenarios.
>
>
> > Q3: The authors mentioned AgentBench in Sec 2 as ... are limited in terms of evaluating LLMs in dynamic environments., but in fact, they have static offline task planing evaluation (Mind2Web).
>
> In this context, "dynamic" specifically refers to an environment that is constantly changing, shaped by the actions of other agents. This description emphasizes the "unpredictability" of the outcomes that agents face after performing an action. That is, the results are determined not only by the agent's own actions but also by the concurrent actions of other agents. In contrast, Mind2Web, while being a complex system, has a relatively predictable environment, as the outcome is constituted by information that websites provide. This differs significantly from the environment in AucArena. In AucArena, due to the uncertainty of other agents' actions, the environment and challenges faced by agents are more dynamic and unpredictable.
>
>
> > Q4: On the variety of bidding items.
>
> The variety of bidding items is not the primary concern of our simulation. Instead, the choice of items valued at $2000 and $10000 is mainly to make the evaluation more controllable. This setup is intended to mimic two common scenarios encountered in reality: products that are inexpensive with limited subsequent markups/profit, and expensive items that may attract more imaginative premiums.
> The rationale for setting these two types of items is to prevent the complication of too many confounding factors at play.

---

> > ### Author Response · Authors · 2023-11-18
> > **Response #2**
> >
> > > Q5: On the presentation of the personal preference/desire.
> >
> > We would like to clarify that they are not conflict with each other. In Sec 3.1, we primarily discuss the modeling of a general auction. The statement ``For simplicity, we do not introduce additional values such as various personal preferences and emotional value of items.``: we indeed do not actually introduce personal perferences for items. The utility function formula below is an example of what motivates a profit-driven bidder.
> >
> > But moving to Sec 3.2, which is Bidder Agent modeling, the statement ``...yet some might also have non-monetary motivations, such as a personal desire to own a specific item.`` This is a general phrasing of the desire of bidder agents in real life, which is true, not what we have implemented in our experiments.
> >
> > In our experiments, we only implemented two types of desires: maximize profit and maximize the number of items. Both of them do not involve personal preference for certain items, but general utility functions for an agent to plan its bidding. In other words, bidders with the same desire will not have different, personal utilities. We will make modifications to our presentation to reduce confusion.
> >
> >
> > > Q6: Participants number in AucAreana
> >
> > Thank you for pointing out this question. We will revise the description in Sec4.3. In the Multi-Agent Intraspecific Competition setting, each auction involves the participation of **six** agents: one rule-based, one PaLM-2, one Claude-1.2, one Claude-2, one GPT-3.5, and one GPT-4. All LLM agents share the same goal of maximizing profits. It's akin to placing different LLMs in a game arena, where they compete against each other to see which one demonstrates stronger capabilities (as shown in TrueSkill in Figure 5).
> >
> > > Q7: The average profit of Claude-1.2/2 is higher than GPT-4
> >
> > To facilitate the auction process, we corrected the belief errors in the models. As stated, ``Note that we correct their belief errors for future bidding, so we can focus on evaluating more advanced strategic abilities.`` In other words, we have decoupled two abilities. Figure 2 places greater emphasis on the model's ability to monitor the auction environment, while Figure 3 examines the impact of whether the planning is adaptive to average profit when given accurate information.

---

> ### Comment · Reviewer_TdM7 · 2023-11-21
>
> Thank you for the detailed response.
>
> My question on (Q7) the difference between Corrected Failure Rate and average profit becomes clear now. Here are the remaining comments:
>
> **(Q1, Q2, Q3, Q4)**
>
> After reading the response, I still concern that it is unclear what aspects of language model agents would be evaluated through this benchmark. For instance, the authors mention that `"dynamic" specifically refers to an environment that is constantly changing, shaped by the actions of other agents. This description emphasizes the "unpredictability" of the outcomes that agents face after performing an action`. However, other environments solved with LLMs, such as household simulators, web/computer control, MineCraft, robotic navigation, are also equally "unpredictable" for LLMs because of their open-endedness (LLMs cannot know all the possible consequence in advance). Also, I wonder if enough diverse situations happen in AucAreana to satisfy the authors arguments. In fact, (1) this simulator only has two items ($2000, $10000), (2) except for multi-agent (i.e. multi LLM) settings, the other participants are rule-based agents, (3) as far as checking Figure 5, different LLMs often achieves the similar TrueSkill Scores. If the possible situations are not diverse, we cannot test the LLM's ability of adaptive strategic thoughts.  If you write tldr on the insights, that might be "GPT-4 works well and Claude sometimes well"?  Even compared to other agentized LLM works, I'm not fully sure the novel insights from the paper.
>
>
>
>
> **(Q5, Q4)**
>
> Because the revision is allowed in this period, the update of paper would be expected. For Question 5, it is quite confusing to mix the proposed methods (based on BDI) and common belief (participants have their own desire).

---

> > ### Author Response · Authors · 2023-11-22
> >
> > Thank you for your insightful feedback!
> >
> > We appreciate the opportunity to further elucidate the unique aspects of our AucArena environment and the strategic capabilities it assesses in language model agents. A key distinction of our environment lies in its emphasis on long-term strategic planning. Agents in arena are not merely reacting to immediate stimuli; they are required to effectively manage resources (e.g., save money for future bids), anticipate future competitive dynamics (e.g., reason about states where their plans are not carried out due to losing bids), and adapt their strategies accordingly. This focus on forward-thinking and resource reallocation is a critical element that sets our benchmark apart from other environments.
> >
> > The environment's design allows for the easy introduction of more diverse parameters such as item types, bidder objectives, etc., offering a *scalable and customized platform* for future studies to increase complexity and examine agent behaviors under a wider range of conditions. As for the diversity of scenarios in AucArena, this choice of limited item types was deliberate to facilitate a more controlled and analyzable setup. Yet, according to the Interspecific Competition experiment in Sec. 4.3, these two representative types of items (cheap vs. expensive) can help us study interesting collective behaviors from agents with different objectives. In the future, we will make the arena more diverse to encourage comparisons and competitions that yield more interesting findings and conclusions.
> >
> > Furthermore, we wish to clarify the experimental design concerning the use of rule-based agents. Rule-based agents are introduced *only* during multi-agent competitions as a baseline. For experiments on rationality (Sec. 4.1) and adaptability (Sec. 4.2), the players in the arena are of the same backbone model. This "self-competition" format ensures that the observed behaviors and strategies are solely attributable to the capabilities of the model under examination, thereby providing a clearer understanding of each model's strategic behaviors.

---

### Official Review · Reviewer_eqke · 2023-10-31

**Soundness:** 2 fair
**Presentation:** 2 fair
**Contribution:** 2 fair
**Rating:** 3
**Confidence:** 4

**Summary:**

The paper proposes AucArena, a simulated auction environment for evaluating the strategic planning and execution abilities of LLMs. The idea of using auctions to assess skills like strategic reasoning, execution, and adaptivity in a multi-agent environment is interesting. With AucArena, the paper also proposes a new LM auction agent: LLMs act as bidders using prompting for actions like planning/ replanning, bidding, and belief updating. Experiments analyze LLM in different tasks in different variations of the environment that test rationality, adaptivity, and competition (with some asymmetry).

**Strengths:**

- The auction simulation is a creative testbed for evaluating LLMs in a dynamic, competitive setting requiring strategic talents.
- The experiments thoroughly probe diverse LLM models in the proposed arena. Testing multiple LLMs provides useful insights into their relative proficiencies.
- Assessing strategic abilities of LLMs is an important open problem and this work makes a valuable first step. The arena concept has promise if further developed.

**Weaknesses:**

- The introduction and problem framing focus heavily on general capabilities, detached from the specific auction setting and contributions. Centering the intro around auctions would improve coherence.
- I would have loved to see a motivation of specifically why agents competing in auctions should be studied where lying and deception could be emergent. Why are auctions special? And not other cooperative settings? A discussion on this is improtant.
- The auction design lacks complexity with only basic factors like item values and bidder budgets. Expanding to more realistic auctions could better evaluate strategic skills. Authors provide suggestions in Appendix C, which if included would greatly improve the paper.
- Prompting details and ablations could provide more insights: Why is the specific prompt design superior to others? Ablations of the prompt with comparisons to other prompts and agents would make the work stronger.
- No discussion on the advantages of an LLM agent over a specialized strategic agent. The choice of LLMs as the agent model (over a symbolic RL agent for example) needs justification.
- Missing Citations: Strategic Reasoning with Language Models [1] seems closely related to the method in the paper with planning, replanning, belief tracking and value estimation of other agents. Other works relating to factoring [2], llm cascades [3], need to be cited and discussed.
- The comparison to human play provides useful context, but the protocol for human evaluation seems ad hoc. Standardizing the process and using skilled human players as a benchmark could make this comparison more rigorous. Increasing the number of human participants could be a good place to start.
- The method was not immediately clear, parts of the instructions could be included in the main paper to make things clearer.
- The evaluation metrics seemed narrow, focusing on the outcomes. It would be great if the authors could translate some of the qualitative measures about strategy into objective numbers.
- Adding an open source model to the benchmark would be interesting!

[1] Gandhi, K., Sadigh, D., & Goodman, N. D. (2023). Strategic Reasoning with Language Models. arXiv preprint arXiv:2305.19165.

[2] A. Stuhlmüller and J. Reppert and L. Stebbing (2022). Factored Cognition Primer. Retrieved December 6, 2022 from https://primer.ought.org.

[3] Dohan, David, et al. "Language model cascades." arXiv preprint arXiv:2207.10342 (2022).

**Questions:**

### Suggestions
Specified in detail in weaknesses, summarized here:
- Improve the introduction and motivation, being specific to auctions and using LLMs with auctions.
- Enrich the auction design with more realism and complexity as outlined in the paper (App C).
- Ablate prompting methodology, compare to stronger baselines.
- Increase and formalize human evaluations. Compare to skilled players.
- Discuss how the approach builds on related work in strategic, factored LLMs.

---

> ### Author Response · Authors · 2023-11-18
> **Response #1**
>
> Thank you for your careful reading of our paper and your valuable suggestions! We will revise the paper as suggested. Please find our responses to your questions below.
>
> > Motivation for Choosing Auctions
>
> The major idea behind our paper is to activate and evaluate the potential and abilities of agents powered by LLMs for tasks that require strategic planning and resource management, which are general but critical in many scenarios like auctions. In the original manuscript, we discuss these general capabilities of LLMs in the context of dynamic environments. We will revise the introduction to more directly link these capabilities to our specific auction setting, and outline the unique challenges and opportunities presented by auction environments for LLMs.
>
> The rationale for focusing on auctions, as opposed to other cooperative settings, is grounded in the unique strategic complexities auctions offer. Moreover, the convenience in quantifying its outcomes, such as total profit or number of items won, can give us clear ranks to the competitors in this arena, showing their differences in terms of these abilities. Since our auction arena is quite extensible, in future work, we will add discussion and negotiation sessions to the auction, with a specific focus to analyze deceptive and cooperative abilities of LLM agents.
>
> > Auction Design Complexity
>
> Thank you for pointing this out. It is easy to change the setting of our auction arena by setting different parameters. However, we deliberately reduce the complexity of the auction design to facilitate analysis (which is difficult in researches on simulations), otherwise too many moving parts will add unnecessary confounders to the results. One benefit from the auction arena is that we can control the setting of the arena according to the ability we want to test LLM agents. For example, as shown in the paper, we can change the objective that drives an agent, or change the number and price of items to create more demanding and long-term auctions.
>
> > Prompting Methodology Ablations
>
> Thank you for the suggestions. Our prompt basically shares the idea of ReAct [1] and Reflexion [2], which asks the model to reason before taking any actions, and then asks the model to reflect on the results and update for upcoming events. We have done ablation studies on the modular level, such as taking out planning/replanning/learning modules from the system to see their effects. We didn't do specific wording-level ablation study because 1) every model in our experiments shares the same, working prompt, so a fair comparison can be maintained, and 2) it's a little expensive to spend our limited API credits on such experiments while we can focus on so many intriguing possibilities from auction arena.
>
> > Justification for Using LLM Agents
>
> Compared with language agents, our choice of LLM-powered agents is underpinned by their demonstrated flexibility and (zero-shot) generalizability in complex, language-based tasks. LLMs, with their ability to process, reason, and generate human-like language, are particularly suited for the nuanced decision-making required in many real-life scenarios. Consequently, agents powered by LLM demonstrate superior capabilities in addressing real-world tasks compared to previous specialized RL agents. These RL agents typically necessitate extensive training within specific environments and often struggle to adapt to communicative contexts during multi-agent interactions. Importantly, the challenge lies in feasibly training these RL agents in such environments, particularly given the absence of coherent training data. On the other hand, our LLMs are zero-shot agents, thus eliminating this issue. Therefore, it is imperative to understand and analyze their abilities before we actually put these LLM agents under complicated, high-stake situations.
>
> > Evaluation Metrics
>
> Thank you for the suggestion! Quantifiable metrics is an advantage of our auction arena, compared with some other agent simulations. We have the following quantifiable metrics, which are not only about outcomes:
> 1) final performance, e.g., total profit for profit-driven bidders, number of items for item-driven bidders,
> 2) measuring bidding rationality, i.e., Corrected Failure Rate, recording belief errors,
> 3) behavioral faithfulness, i.e., correlation between executions and plans,
> 4) performance ranking between multiple agents, i.e., TrueSkill scores.
>
> Although we agree that further digging into strategies such as intermediate reasoning process could be interesting, it is difficult to objectively do so other than manual check or case study, and using LLMs as evaluators creates unintentional biases, such as GPT-4 favors texts generated by GPT-4.
>
> [1] Yao, Shunyu, et al. "React: Synergizing reasoning and acting in language models." In ICLR 2023.
>
> [2] Shinn, Noah, et al. "Reflexion: Language agents with verbal reinforcement learning." In NeurIPS 2023.

---

> ### Author Response · Authors · 2023-11-18
> **Response #2**
>
> > Adding an Open Sourced Model
>
> That's a great suggestion! We add an open source model, [OpenChat-3.5 7B](https://huggingface.co/openchat/openchat_3.5), which proves to be better even than ChatGPT (GPT-3.5) on some benchmarks. We add it into the intra-specific multi-agent competition, with PaLM-2, GPT-3.5, GPT-4, Claude-2.0, Claude-Instant-1.2, and Rule bidder as fellow bidders. The setting is the same as that of Sec 4.3, except we have 7 bidders instead of 6. We repeat the experiments by 20 times, and the average results are as follows. The results shows that OpenChat-3.5 is at least good at numerical skills and processing facts, as shown by its less belief errors (according to CFR scores) than GPT-3.5 and Palm-2.
>
> | Model | # Engagement | CFR (% Failed Bids) | CFR (% Self Belief Error) | CFR (% Other Belief Error) | # Items Won | Money Left ($) | Profit ($) |
> |-------|------------------|-----------------|-----------------|------------------|-----------|------------|--------|
> | Rule | 24.86 | - | - | - | **4.86** | 425.0 | 3425.0 |
> | GPT-3.5 | 6.86 | **0.18** | 12.70 | 13.80 | 1.04 | 6840.9 | 1840.9 |
> | GPT-4 | 25.5 | 1.09 | **3.68** | **0.90** | 3.27 | 953.4 | 4044.3 |
> | Claude-2 | 20.95 | 6.36 | 1.77 | 4.04 | 2.54 | 2131.8 | **4131.8** |
> | Claude-1.2 | 14.41 | 10.54 | 11.30 | 8.09 | 1.95 | 2786.4 | 3240.9 |
> | Palm-2 | 24.77 | 17.45 | 10.09 | 12.80 | 4.68 | 654.5 | 3654.5 |
> | OpenChat-3.5 | 6.5 | 2.32 | 9.50 | 11.40 | 0.68 | 7750.0 | 931.8 |
>
> As for the TrueSkill score, there is still plenty room for improvement for OpenChat-3.5 in such a dynamic competition.
>
> | Model | $\mu$ | $\sigma$ |
> |-------|-----|--------|
> | Claude-2 | 28.4392 | 1.1258 |
> | GPT-4 | 26.8398 | 1.1187 |
> | Palm | 26.6751 | 1.0832 |
> | Rule | 26.3237 | 1.0710 |
> | Claude-1.2 | 25.4109 | 1.0818 |
> | GPT-3.5 | 22.7858 | 1.1307 |
> | OpenChat-3.5 | 17.8331 | 1.2252 |

---

> > ### Comment · Reviewer_eqke · 2023-11-23
> > **Response to authors**
> >
> > I thank the authors for their reply!
> > I am glad that they added an open-source model to the benchmark and found it to be competitive. This certainly makes the work more accessible and replicable.
> > I understand the justification for LLM agents, but I would have liked to see a revised manuscript with the discussion.
> >
> > I will maintain my initial evaluation score. This paper's main contribution is an interesting environment for testing language models. However, I believe that incorporating the features outlined in Appendix C is necessary to realize the full potential of this environment.
> >
> > Best

---

### Official Review · Reviewer_PeRU · 2023-11-10

**Soundness:** 2 fair
**Presentation:** 2 fair
**Contribution:** 2 fair
**Rating:** 3
**Confidence:** 5

**Summary:**

This paper studies an interesting problem about the LLM's ability in auction, which can test whether the LLM can obtain high reward. The authors designed some prompts, and the results show LLMs are still limited.

**Strengths:**

1. The studied problem is quite interesting.
2. The paper is written well.
3. The appendix provides enough details for the prompts.

**Weaknesses:**

1. The technical contribution is limited. The fundamental mechanism for the LLM agent is quite simple.
2. The results are not so promising. There is no sufficient comparison among different LLMs.
3. The setting of the auction is quite simple. I suggest the authors test different settings (those standard ones), and try to provide more insightul conclusions, compared with human.

**Questions:**

Please address my concerns in Weaknesses.

**Details Of Ethics Concerns:**

There is no ethics concern.

---

> ### Author Response · Authors · 2023-11-18
>
> Thanks for your comments. Please find our responses to your questions below.
>
> > Q1: The technical contribution is limited. The fundamental mechanism for the LLM agent is quite simple.
>
> Our prompting strategy seems simple, but our zero-shot CoT-style prompting is consistent with the current SOTA for working with these models, and novel in that we are applying it to dynamic scenarios.
>
> Furthermore, the design of Agents in the auction context is NOT simple. This is due to the many changes in the situation as the auction progresses. Key processes like Belief Update, Replanning, and Learning from Past Experiences are highlighted as areas where LLM Agents are expected to improve toward greater intelligence.
>
> > Q2:  The results are not so promising. There is no sufficient comparison among different LLMs.
>
> For clarification, we note that our experiments involve an exhaustive set of the current best LLMs, namely PaLM-2, Claude-1.2, Claude-2, ChatGPT and GPT-4, all studied individually (Sec 4.1 & 4.2), and in combination and competition with one another (Sec. 4.3) and against human agents (Fig 6). In our view, this makes our studies fairly comprehensive, with 7 experiments.
>
> Regarding the results, we do not take the view that ``The results are not so promising``, in the absence of more details here we are unsure what you mean. On the contrary, while our results are nuanced, our best models such as GPT4 do indeed exhibit considerable skills in the auction arena, such as the ability to maintain coherent belief states (Fig 2), generate rational long-term plans (Fig 4) that are highly correlated with their behavior (Tab 1) and ultimately behave in a way that achieves their intended goals (Fig 3 & 5). Of course, there is still variability in the skills of different models, and models are indeed not perfect and have much room for improvement; this, in our view, is the main value of our simulation environment and our work.
>
>
> > Q3: The setting of the auction is quite simple. I suggest the authors test different settings (those standard ones), and try to provide more insightul conclusions, compared with human.
>
> Our auction setting closely standard the open ascending price auction model. In the absence of more details here, we are not sure what the reviewer means by ``standard scenario``.
>
> Additionally, we have tried different settings in our experiments:
> - In Sec. 4.1, two bidders auction for ten items with different prices.
> - In Sec. 4.2, we adjusted the order of the items as a control variable.
> - In Sec. 4.3, we had six different auction agents competing with each other for different items.
> - In the Interspecific Competition experiment, different agents were set with two different types of objectives.
> - In Figure 6, we provided a comparison with human bidding, and we indeed found that humans are not always far ahead in this kind of competitive environment. As a benchmark, our conclusions have already been written in the abstract.

---

### Public Comment · ~Guohao_Li1 · 2023-11-14

The paper introduces AUCARENA, a novel simulation environment designed to evaluate the capabilities of Large Language Models (LLMs) in complex, strategic auction scenarios. The paper focuses on three aspects: the rationality of LLM agents, their adaptability, and their performance in multi-agent competition. The researchers use state-of-the-art LLMs, including GPT-4, as agents in their simulations. The paper's findings suggest that LLMs, through simple prompting, can effectively engage in auctions, manage resources, and adapt to changing environments.

Thanks for the great work. It could also be beneficial to discuss prior work on multi-LLM agents for the study of cooperative settings [1] and what are the unique challenges of multi-agent competition compared with multi-agent cooperation.

[1] Li, Guohao, Hasan Abed Al Kader Hammoud, Hani Itani, Dmitrii Khizbullin, and Bernard Ghanem. "CAMEL: Communicative Agents for" Mind" Exploration of Large Language Model Society." NeurIPS 2023

---

### Author Response · Authors · 2023-11-18
**General Response to All Reviewers**

We appreciate the valuable comments from the reviewers and are encouraged that reviewers found that "The studied problem is quite interesting", that our experiments "thoroughly probe diverse LLM models...[that] provides useful insights into their relative proficiences" and that "the paper is clearly written and easy to follow"

We'd like to address two primary issues raised: 1) the choice of simulation as a
testbed for LLM agent development, and 2) the specific selection of an auction
environment.

> Why Simulations for LLM Agent Development?

We believe simulations are to agents currently what datasets are to NLP previously: we need dynamic simulations to evaluate actionable agents. Just like we need different NLP datasets to test different abilities of models/LLMs, we need more types of simulations that reflects different abilities for agents before putting them into real world. Simulations, as opposed to static NLP datasets, offer dynamic, unpredictable environments that are crucial for testing the strategic reasoning and adaptability of LLMs in real-world scenarios. Traditional NLP benchmarks often assess agents in static contexts, which do not sufficiently probe the agent's ability to formulate and revise long-term plans and strategies. Our auction simulation, AucArena, serves as a realistic yet controlled environment where LLM agents can demonstrate their strategic reasoning, resource and risk management skills, crucial in competitive settings.

> Why an Auction-Based Simulation?

Auctions present a strategic, complex, and quantifiable microcosm of real-world economic and social interactions. This setting enables us to objectively measure agent capabilities using clear metrics like total profit and bidding success rates. The versatility of auction simulations allows for the exploration of a range of behaviors and interactions, targeting specific strategic angles crucial in real-world scenarios. While other simulations are indeed valuable, the auction environment specifically focuses on dynamic decision-making and competitive interactions, offering a unique perspective on LLM capabilities.

We believe that the choice of an auction simulation complements other forms of agent evaluation and contributes uniquely to the broader understanding of LLMs in dynamic and competitive environments. The insights garnered from AucArena will not only advance our understanding of LLM agents but also pave the way for future research in AI.

---

### Comment · Area_Chair_6a9e · 2023-11-20
**Please engage in reviewer-author discussions**

Reviewers - I encourage you to read the authors' response carefully and let the authors know whether their response has addressed your comments.